# Coneheads: Hierarchy Aware Attention

**Albert Tseng**
Cornell University
albert@cs.cornell.edu

**Tao Yu**
Cornell University
tyu@cs.cornell.edu

**Toni J.B. Liu**
Cornell University
jl3499@cornell.edu

**Christopher De Sa**
Cornell University
cdesa@cs.cornell.edu

## Abstract

Attention networks such as transformers have achieved state-of-the-art performance in many domains. These networks rely heavily on the dot product attention operator, which computes the similarity between two points by taking their inner product. However, the inner product does not explicitly model the complex structural properties of real world datasets, such as hierarchies between data points. To remedy this, we introduce cone attention, a drop-in replacement for dot product attention based on hyperbolic entailment cones. Cone attention associates two points by the depth of their lowest common ancestor in a hierarchy defined by hyperbolic cones, which intuitively measures the divergence of two points and gives a *hierarchy aware* similarity score. We test cone attention on a wide variety of models and tasks and show that it improves task-level performance over dot product attention and other baselines, and is able to match dot-product attention with significantly fewer parameters. Our results suggest that cone attention is an effective way to capture hierarchical relationships when calculating attention.

## 1 Introduction

In recent years, attention networks have achieved highly competitive performance in a variety of settings, often outperforming highly-engineered deep neural networks [5, 8, 31]. The majority of these networks use dot product attention, which defines the similarity between two points $u, v \in \mathbb{R}^d$ by their inner product $u^\top v$ [31]. Although dot product attention empirically performs well, it also suffers from drawbacks that limit its ability to scale to and capture complex relationships in large datasets [33, 27]. The most well known of these issues is the quadratic time and memory cost of computing pairwise attention. While many works on attention mechanisms have focused on reducing the computational cost of dot product attention, few have considered the properties of the dot product operator itself [33, 6].

Many real world datasets exhibit complex structural patterns and relationships which may not be well captured by an inner product [30, 16]. For example, NLP tasks often contain hierarchies over tokens, and images may contain clusters over pixels [34, 14]. Motivated by this, we propose a new framework based on hyperbolic entailment cones to compute attention between sets of points [10, 37]. Our attention mechanism, which we dub "cone attention", utilizes partial orderings defined by hyperbolic cones to better model hierarchical relationships between data points. More specifically, we associate two points by the depth of their lowest common ancestor (LCA) in the cone partial ordering, which is analogous to finding their LCA in a latent tree and captures how divergent two points are.

Cone attention effectively relies on two components: hyperbolic embeddings and entailment cones. Hyperbolic embeddings, which use the underlying geometric properties of hyperbolic space, give low-

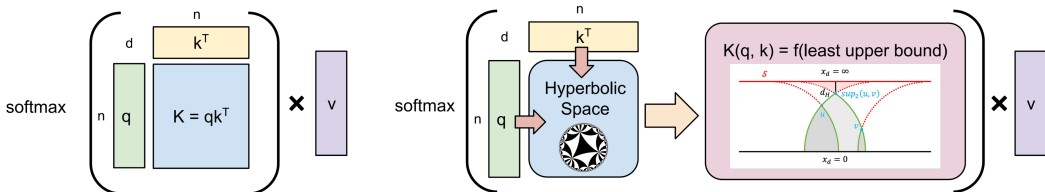

Figure 1: Overview of cone attention vs. dot product attention. In dot product attention (left), similarity scores are calculated with $K = qk^\top$. In cone attention (right), $q$ and $k$ are first projected onto hyperbolic space. Then, pairwise similarity is calculated from the lowest common ancestor of points in the partial ordering defined by entailment cones. Cone attention allows us to explicitly encode notions of hierarchy in attention, and empirically gives better performance than dot product attention.

distortion embeddings of hierarchies that are not possible with Euclidean embeddings [25]. Entailment cones, which rely on geometric cones to define partial orders between points, allow us to calculate explicit relationships between points, such as their LCA [10, 37]. To the best of our knowledge, we are the first to define a hierarchy-aware attention operator with hyperbolic entailment cones.

Functionally, cone attention is a drop-in replacement for dot product attention. We test cone attention in both "classical" attention networks and transformers, and empirically show that cone attention consistently improves end task performance across a variety of NLP, vision, and graph prediction tasks. Furthermore, we are able to match dot product attention with significantly fewer embedding dimensions, resulting in smaller models. To summarize, our contributions are:

- We propose cone attention, a hierarchy aware attention operator that uses the lowest common ancestor of points in the partial ordering defined by hyperbolic entailment cones.

- We evaluate cone attention on NLP, vision, and graph prediction tasks, and show that it consistently outperforms dot product attention and other baselines. For example, we achieve +1 BLEU and +1% ImageNet Top-1 Accuracy on the `transformer_iwslt_de_en` and DeiT-Ti models, respectively.

- We test cone attention at low embedding dimensions and show that we can significantly reduce model size while maintaining performance relative to dot product attention. With cone attention, we can use 21% fewer parameters for the IWSLT'14 De-En NMT task.

## 2 Background

In this section, we provide background on attention mechanisms, motivate the use of hyperbolic space to embed hierarchies, and describe our choice of entailment cones to encode partial orderings.

### 2.1 Attention

The attention operator has gained significant popularity as a way to model interactions between sets of tokens [31]. At its core, the attention operator $A$ performs a "lookup" between a single query $q_i \in \mathbb{R}^d$ and a set of keys $k \in \mathbb{R}^{n \times d}$, and aggregates values $v \in \mathbb{R}^{n \times d}$ associated with the keys to "read out" a single value for $q_i$. Mathematically, this can be represented as

$$A(q_i, k) = C \sum_j \left( K(q_i, k_j) v_j \right) \tag{1}$$

where $C \sum_j K(q_i, k_j) = 1$. In "traditional" dot product attention, which has generally superseded "older" attention methods such as Additive Attention and Multiplicative Attention [4, 15],

$$K(q_i, k_j) = \exp\left(\frac{q_i k_j}{\sqrt{d}}\right) \qquad C = \frac{1}{\sum_j K(q_i, k_j)} \qquad A(q_i, k) = \text{softmax}\left(\frac{q_i k^\top}{\sqrt{d}}\right) v \tag{2}$$

similarity is scored with a combination of cosine similarity and embedding magnitudes.

Existing works have proposed replacing dot product attention to various degrees. A large body of works focus on efficiently computing dot product attention, such as with Random Fourier Features and low-rank methods [23, 33]. These methods generally perform worse than dot product attention, as they are approximations [38]. Some recent works, such as EVA attention, parameterize these approximations and effectively get a larger class of dot-product-esque attention methods [38]. Beyond this, Tsai et al. [29] replace $K$ with compositions of classical kernels. Others extend dot product attention, such as by controlling the "width" of attention with a learned Gaussian distribution or by using an exponential moving average on inputs, although extensions do not usually depend on $K$ [12, 16]. Closer to our work, Gulcehre et al. [11] introduced hyperbolic distance attention, which defines $K(q_i, k_i) = \exp(-\beta d_\mathbb{H}(q_i, k_i) - c)$ where $d_\mathbb{H}$ is the hyperbolic distance, $\beta \in \mathbb{R}^+$, and $c \in \mathbb{R}$. $K$ can be interpreted as an analog of the distance from $q_i$ to $k_i$ on a latent tree. Finally, in an orthogonal direction, Tay et al. [26] ignore token-token interactions and synthesize attention maps directly from random alignment matrices. Whether token-token interactions are actually needed is outside the scope of this work, and we compare cone attention accordingly.

## 2.2 Hyperbolic Space

$d$-dimensional Hyperbolic space, denoted $\mathbb{H}_d$, is a simply connected Riemannian manifold with constant negative sectional curvature [2]. This negative curvature results in geometric properties that makes hyperbolic space well-suited for embedding tree-like structures [25, 35]. For example, the volume of a hyperbolic ball grows exponentially with respect to its radius; in a tree, the number of leaves grows exponentially with respect to depth. Furthermore, $d_\mathbb{H}(u, v) \approx d_\mathbb{H}(u, O) + d_\mathbb{H}(O, v)$, where $O$ is the origin, which again mirrors a tree where $d_T(u, v) = d_T(u, \text{LCA}(u, v)) + d_T(\text{LCA}(u, v), v)$. Since hyperbolic space cannot be isometrically embedded into Euclidean space, it is usually represented on a subset of Euclidean space by a "model" of $\mathbb{H}_d$ [11]. These models are isometric to each other, and the key differences between them lie in their different parameterizations, which allow for cleaner computations and visualizations for certain tasks.

In this work, we primarily use the Poincaré half-space model, which is the manifold $\mathsf{H}^d = (\mathcal{U}^d, g_u)$ where $\mathcal{U}^d = \{x \in \mathbb{R}^d : x_d > 0\}$, $g_u(x) = g_e/x_d^2$, and $g_e$ is the Euclidean metric [2]. In the Poincaré half-space, "special" points and curves have particularly nice Euclidean forms. Ideal points, or points at infinity, are the points where $x_d = 0$ (the "$x$-axis") and the single point $x_d = \infty$ at which all lines orthogonal to the $x$-axis converge. Geodesics, the shortest path between two points, are Euclidean semicircles with the origin on the $x$-axis or vertical rays orthogonal to the $x$-axis. Horospheres, curves where all normal curves converge at the ideal point, are represented by either an Euclidean ball tangent to the $x$-axis or a horizontal hyperplane when the ideal point is $x_d = \infty$.

## 2.3 Entailment Cones

Entailment cones in hyperbolic space were first introduced by Ganea et al. [10] to embed partial orders. The general concept of Ganea's entailment cones is to capture partial orders between points with membership relations between points and geodesically convex cones rooted at said points. That is, if $u \in$ the cone of $v$, then $v \prec u$. Ganea's cones (figure 2 right) are defined on the Poincaré ball by a radial angle function $\psi(r)$, with an $\epsilon$-ball around the origin where cones are undefined [2, 10]. This makes learning complicated models with Ganea's cones difficult, as optimization on the Poincaré ball is nontrivial and the $\epsilon$-ball negatively impacts embedding initializations [37, 10].

In this work, we instead use the shadow cone construction introduced in [37] and operate on the Poincaré half-space, which makes computing our desired attention function numerically simpler. Shadow cones are defined by shadows cast by points and a single light source $\mathcal{S}$, and consist of the penumbral and umbral settings (figure 2 left quadrant). In the penumbral setting, $\mathcal{S}$ is a ball of fixed radius and points are points. The shadow and cone of $u$ are both the region enclosed by geodesics through $u$ tangent to $\mathcal{S}$. In the umbral setting, $\mathcal{S}$ is instead a point, and points are centers of balls of fixed radius. Here, the shadow of $u$ is the region enclosed by geodesics tangent to the ball around $u$ that intersect at $\mathcal{S}$. However, to preserve transitivity, the cone of $u$ is a subset of the shadow of $u$ (see figure 2). The shadow cone formulation can also be achieved with subset relations between shadows instead of membership relations between points and cones, which may be conceptually clearer.

**Infinite-setting Shadow Cones.** For penumbral cones, when $\mathcal{S}$ is a ball with center at $x_d = \infty$, $\mathcal{S}$'s boundary is a horosphere of user-defined height $h$. Here, all shadows are defined by intersections of

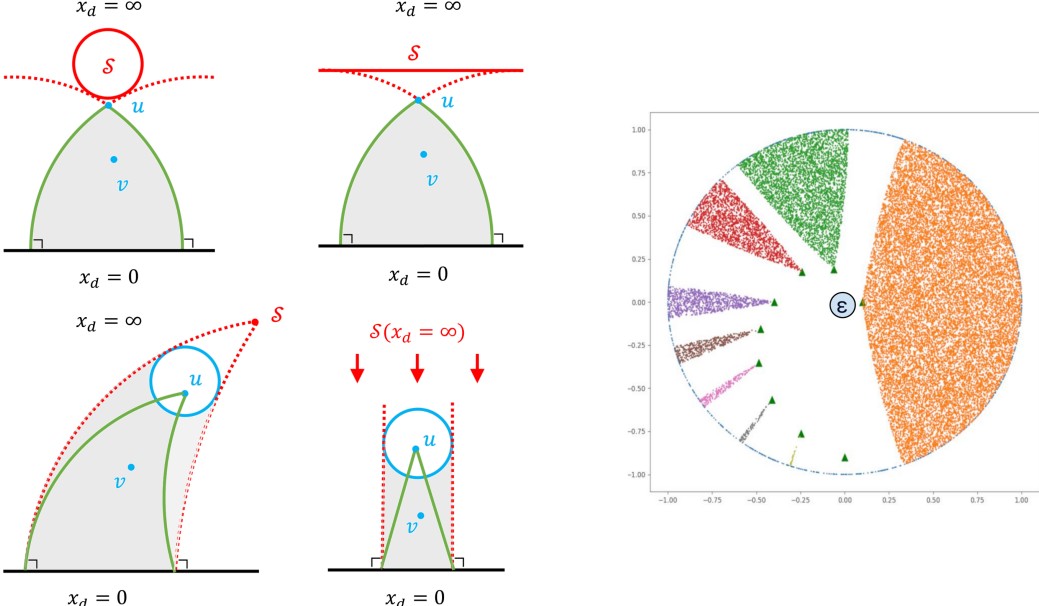

Figure 2: (Left Quadrant) Clockwise from top left: finite setting penumbral cone, infinite setting penumbral cone, infinite setting umbral cone, and finite setting umbral cone. All figures are in $\mathsf{H}^2$. Shadows are represented by shaded regions, and cones are enclosed in green. In all figures, $u \prec v$ since $v$ is in the cone of $u$. (Right) Ganea's entailment cones, as defined on the Poincaré ball. Note the $\epsilon$-ball where cones are not defined, which makes optimization nontrivial. Figure from [10].

Euclidean semicircles of Euclidean radius $h$. Notably, the infinite setting penumbral cone construction is similar to Ganea's cones under an isometry from the Poincaré half-space to the Poincaré ball where $\mathcal{S}$ maps to the $\epsilon$-ball [37]. For umbral cones, when $\mathcal{S}$ is $x_d = \infty$, shadows are regions bounded by Euclidean lines perpendicular to the $x$-axis.

Unlike Ganea's cones and penumbral cones, umbral cones are not geodesically convex [37]. That is, the shortest path between two points in an umbral cone may not necessarily lie in the cone. In a tree, this corresponds to the shortest path between two nodes not being in the subtree of their LCA, which is not possible. Empirically, while still better than dot product attention, umbral attention usually performs worse than penumbral attention.

## 2.4 Lowest Common Ancestor (LCA) and Least Upper Bound

The LCA of two nodes $u, v$ in a directed acyclic graph is the lowest (deepest in hierarchy) node that is an ancestor of both $u$ and $v$. Although the two terms are similar, the LCA is *not* the same as the least upper bound of a partial ordering. The least upper bound of two points $x, y$ in a partial ordering, denoted $\sup(x, y)$, is the point $p$ such that $p \preceq x, y$ and $\forall q$ where $q \preceq x, y$, $q \preceq p$. The key difference is that all other upper bounds must precede the least upper bound, while not all ancestors of $u$ and $v$ must also be ancestors of $\mathrm{LCA}(u, v)$. Furthermore, $p$ may not actually exist.

## 3 Hierarchy Aware Attention

Here, we describe cone attention using the shadow cone construction, and discuss projection functions onto $\mathbb{H}^d$ that allow us to use cone attention within attention networks. All definitions use the infinite-setting shadow cones, and proofs and derivations are in the appendix. For clarity, we refer to Ganea's entailment cones as "Ganea's cones" and the general set of cones that captures entailment relations (e.g. Ganea's cones and shadow cones) as "entailment cones" [10, 37]. Our methods are agnostic entailment cone choice, and can also be used with Ganea's cones.

### 3.1 Cone Attention

We wish to associate $u, v \in \mathsf{H}^d$ by their LCA in some latent tree $T$, which is analogous to finding their LCA, denoted $\sup_2(u, v)$, in the partial ordering defined by entailment cones. Formally,

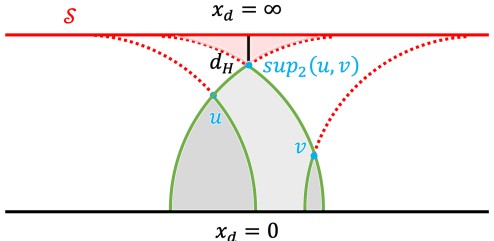

Figure 3: In this example using penumbral cones in $\mathsf{H}^d$, $\sup_2(u, v)$ is the lowest common ancestor of $u$ and $v$. The red region is the set of points $P$ s.t. $p \in P \preceq u, v$. Of these, $\sup_2(u, v)$ is the lowest point whose cone (light gray) contains both $u$ and $v$. Here, $\mathcal{S}$ is the root of all hierarchies, and points closer to $x_d = 0$ are closer to the "leaves" of hierarchies. If $d = 2$, then $\sup_2(u, v)$ is also the least upper bound of $u$ and $v$ in the partial ordering defined by entailment cones.

$$\sup_2(u, v) = r \left( \arg\max_{C:u,v \in C} d_H(\mathcal{S}, r(C)) \right) \tag{3}$$

where $r(C)$ denotes the root of a cone $C$. This corresponds to finding the cone that is farthest away from $\mathcal{S}$, which is the root of all hierarchies in the shadow cones construction. When $d = 2$, $\sup_2(u, v)$ also has the nice property of being the least upper bound $\sup(u, v)$ in the partial ordering defined by shadow cones. Then, we define the similarity between $u$ and $v$ as

$$K(u, v) = f(d_{\mathbb{H}}(\sup_2(u, v), \mathcal{S})) \tag{4}$$

where $f$ is a user-defined monotonically increasing function. If $K(u, v) > K(u, w)$, then $d_{\mathbb{H}}(\sup_2(u, v), \mathcal{S}) > d_{\mathbb{H}}(\sup_2(u, w), \mathcal{S})$, which implies that $u$ and $v$ have a more recent "ancestor" than $u$ and $w$. Thus, $K$ gives a higher similarity score to points who have a more recent "ancestor" in $T$. In the infinite-setting shadow cone construction, $\sup_2(u, v)$ is root of the minimum height (literal lowest) cone that contains both $u$ and $v$, or $\sup_2(u, v) = r \left( \arg\min_{C:u,v \in C} r(C)_d \right)$.

Using this, we provide definitions for $K(u, v)$ in the infinite-setting shadow cone construction. Both the umbral and penumbral definitions correspond to the Euclidean height of $\sup_2(u, v)$. In the penumbral setting, when $\sup_2(u, v)$ does not exist, we return the Euclidean height of lowest light source where $\sup_2(u, v)$ exists. $x_d$ denotes the last dimension of $x$, and $x_{:-1}$ denotes the first $d - 1$ dimensions of $x$. $\gamma \in \mathbb{R}^+$ corresponds to the softmax "temperature" in attention. In the penumbral setting, $r \in \mathbb{R}^+$ is the user-defined height of the horosphere light source. In the umbral setting, $r \in \mathbb{R}^+$ is the user-defined radius of the ball centered at each point.

**Definition 1.** *Penumbral Attention:*

$$K(u, v) = \exp\left( -\gamma \max\left( u_d, v_d, \sqrt{r^2 - \left( \frac{\sqrt{r^2 - u_d^2} + \sqrt{r^2 - v_d^2} - \|u_{:-1} - v_{:-1}\|}{2} \right)^2} \right) \right) \tag{5}$$

*when there exists a cone that contains u and v, and*

$$K(u, v) = \exp\left( -\gamma \sqrt{\left( \frac{\|u_{:-1} - v_{:-1}\|^2 + u_d^2 - v_d^2}{2\|u_{:-1} - v_{:-1}\|} \right)^2 + v_d^2} \right) \tag{6}$$

*otherwise. There exists a cone that contains u and v when*

$$\left( \|u_{:-1} - v_{:-1}\| - \sqrt{r^2 - u_d^2} \right)^2 + v_d^2 < r^2 \tag{7}$$

**Definition 2.** *Umbral Attention:*

$$K(u, v) = \exp\left( -\gamma \max\left( u_d, v_d, \frac{\|u_{:-1} - v_{:-1}\|}{2\sinh(r)} + \frac{u_d + v_d}{2} \right) \right) \tag{8}$$

These definitions possess a rather interesting property – when $u_d = v_d$ and $K$ is normalized across a set of $v$s, cone attention reduces to the Laplacian kernel $K(u_{:-1}, v_{:-1}) = \exp(-\gamma\|u_{:-1} - v_{:-1}\|)$. Since Euclidean space is isomorphic to a horosphere, and $u$ and $v$ are on the same horosphere if $u_d = v_d$, cone attention can also be seen as an extension of the Laplacian kernel [19].

Cone attention and dot product attention both take $O(n^2 d)$ time to compute pairwise attention between two sets of $n$ $d$-dimensional tokens $q, k \in \mathbb{R}^{n \times d}$ [33]. However, cone attention takes more operations than dot product attention, which computes $qk^\top$. In transformers, our PyTorch cone attention implementations with `torch.compile` were empirically 10-20% slower than dot product attention with `torch.bmm` (cuBLAS)[21] (see section **??**). `torch.compile` is not optimal, and a raw CUDA implementation of cone attention would likely be faster and narrow the speed gap between the two methods [21].

## 3.2  Mapping Functions

Here, we discuss mappings from Euclidean space to the Poincaré half-space. These mappings allow us to use cone attention within larger models. The canonical map in manifold learning is the exponential map $\text{Exp}_x(v)$, which maps a vector $v$ from the tangent space of a manifold $\mathcal{M}$ at $x$ onto $\mathcal{M}$ by following the geodesic corresponding to $v$ [24, 2]. In the Poincaré half-space [36],

$$\text{Exp}_x(v)_{:-1} = x_{:-1} + \frac{x_d}{\|v\|/\tanh(\|v\|) - v_d} v_{:-1} \qquad \text{Exp}_x(v)_d = \frac{x_d}{\cosh(\|v\|) - v_d \sinh(\|v\|)/\|v\|} \tag{9}$$

While $\text{Exp}_x(v)$ is geometrically well motivated, it suffers from numerical instabilities when $\|v\|$ is very large or small. These instabilities make using the exponential map in complicated models such as transformers rather difficult. Using `fp64` reduces the risk of numerical over/underflows, but `fp64` significantly reduces performance on GPUs, which is highly undesirable [1].

Instead, we use maps of the form $(x_{:-1}, x_d) \rightarrow (x_{:-1} f(x_d), f(x_d))$ that preserve the exponential volume of hyperbolic space while offering better numerics for large-scale optimization. Our use of alternatives to $\text{Exp}_x(v)$ follows Gulcehre et al. [11]'s psuedopolar map onto the Hyperboloid. Geometrically, since $n$-dimensional Euclidean space is isomorphic to a horosphere in $\mathbb{H}^{n+1}$, these maps corresond to selecting a horosphere with $f(x_d)$ and then projecting $x_{:-1}$ onto that horosphere [19]. To achieve exponential space as $x_d \rightarrow -\infty$, we use functions $f$ of the form $\exp(\cdot)$.

For the infinite-setting umbral construction, since there is no restriction on where points can go in the Poincaré half-space, we map $x \in \mathbb{R}^d$ to $\mathsf{H}^d$ with $\psi : \mathbb{R}^d \rightarrow \mathsf{H}^d$:

$$\psi(x)_{:-1} = x_{:-1} \exp(x_d) \qquad \psi(x)_d = \exp(x_d) \tag{10}$$

For the infinite-setting penumbral construction, since the mapped point cannot enter the light source at height $h$, we map $x \in \mathbb{R}^d$ to area below the light source with $\xi : \mathbb{R}^d \rightarrow \mathsf{H}^d$

$$\xi(x)_{:-1} = x_{:-1} \frac{h}{1 + \exp(-x)} \qquad \xi(x)_d = \frac{h}{1 + \exp(-x)} \tag{11}$$

While $\xi$ is structurally the sigmoid operation, note that $\text{sigmoid}(x) = \exp(-\text{softplus}(-x))$. Since $\text{softplus}(x) \approx x$ for large values of $x$, $\xi$ preserves the exponential volume properties we seek.

## 4  Experiments

Here, we present an empirical evaluation of cone attention in various attention networks. For each model we test, our experimental procedure consists of changing $K$ in attention and training a new model from scratch. Unless otherwise noted in the appendix, we use the code and training scripts that the authors of each original model released. We assume released hyperparameters are tuned for dot product attention, as these models were state-of-the-art (SOTA) when new.

### 4.1  Baselines

Our main baseline is dot product attention, as it is the most commonly used form of attention in modern attention networks. Additionally, we compare cone attention against Gulcehre et al. [11]'s

hyperbolic distance attention and the Laplacian kernel. To the best of our knowledge, few other works have studied direct replacements of the dot product for attention.

Gulcehre et al. [11]'s original hyperbolic distance attention formulation not only computed the similarity matrix $K$ in the Hyperboloid, but also aggregated values $v$ in hyperbolic space by taking the Einstein midpoint with respect to weights $\alpha$ in the Klein model (see appendix for definitions) [24]. However, the Einstein midpoint, defined as

$$m(\alpha, v) = \sum_i \left( \frac{\alpha_i \gamma(v_i)}{\sum_j \alpha_j \gamma(v_j)} \right) \tag{12}$$

where $\gamma(v) = 1/\sqrt{1 - \|v\|^2}$, does not actually depend on how $\alpha$ is computed. That is, $\alpha$ could be computed with Euclidean dot product attention or cone attention and $m(\alpha, v)$ would still be valid. The focus of our work is on computing similarity scores, so we do not use hyperbolic aggregation in our experiments. We test hyperbolic distance attention using both Gulcehre et al. [11]'s original pseudopolar projection onto the Hyperboloid model and with our $\psi$ map onto the Poincaré half-space.

## 4.2 Models

We use the following models to test cone attention and the various baselines. These models span graph prediction, NLP, and vision tasks, and range in size from a few thousand to almost 250 million parameters. While we would have liked to test larger models, our compute infrastructure limited us from feasibly training billion-parameter models from scratch.

**Graph Attention Networks.** Graph attention networks (GATs) were first introduced by Veličković et al. [32] for graph prediction tasks. GATs use self-attention layers to compute node-level attention maps over neighboring nodes. The original GAT used a concatenation-based attention mechanism and achieved SOTA performance on multiple transductive and inductive graph prediction tasks [32]. We test GATs on the transductive Cora and inductive multi-graph PPI datasets [17, 13].

**Neural Machine Translation (NMT) Transformers.** Transformers were first applied to NMT in Vaswani et al. [31]'s seminal transformer paper. We use the fairseq `transformer_iwslt_de_en` architecture to train a German to English translation model on the IWSLT'14 De-En dataset [20, 9]. This architecture contains 39.5 million parameters and achieves near-SOTA performance on the IWSLT'14 De-En task for vanilla transformers [20]. As this model is the fastest transformer to train out of the tested models, we use it for ablations.

**Vision Transformers.** Vision transformers (ViT) use transformer-like architectures to perform image classification [8]. In a ViT, image patches are used as a tokens in a transformer encoder to classify the image. We use the *Data Efficient* Vision Transformer (DeiT) model proposed by FAIR, which uses a student-teacher setup to improve the data efficiency of ViTs [28]. DeiTs and ViTs share the same architecture, and the only differences are how they are trained and the distillation token. We train DeiT-Ti models with 5 million parameters on the ImageNet-1K dataset for 300 epochs[28, 7]. We also train cone and dot product attention for 500 epochs, as we observed that training for more iterations improves performance.

**Adaptive Input Representations for Transformers.** Adaptive input representations were introduced by Baevski and Auli for transformers in language modeling tasks [3]. In adaptive inputs, rarer tokens use lower dimensional embeddings, which serves as a form of regularization. We use the fairseq `transformer_lm_wiki103` architecture (246.9M parameters) and train models on the WikiText-103 language modeling dataset with a block size of 512 tokens [20, 18]. We also test the same architecture without adaptive inputs, which has 520M parameters. This version converges in significantly fewer iterations, allowing us to train such a large model.

**Diffusion Transformers.** Diffusion transformers (DiT) are diffusion models that replace the U-Net backbone with a transformer [22]. DiTs operate on latent space patches, so we expect there to be less hierarchical information vs. taking image patches. We use DiTs to test cone attention when the data is less hierarchical. We train DiT-B/4 models with 130M parameters on ImageNet-1K [22, 7].

Table 1: Performance of various attention methods across models and tasks. $^*$ indicates the model failed to converge or ran into NaN errors. ↑ indicates higher is better, and ↓ indicates lower is better. Cone attention methods (†) generally outperform dot product attention and other baselines. Model default refers to a model's default attention method, which is dot product attention except for GATs.

| Method | NMT IWLST (BLEU ↑) | DeiT-Ti Imagenet Top-1 / 5 (Acc. ↑) 300 Epochs | DeiT-Ti Imagenet Top-1 / 5 (Acc. ↑) 500 Epochs | GAT Cora / PPI (Acc. ↑) |
|---|---|---|---|---|
| Model Default | 34.56 | 72.05 / 91.17 | 73.65 / 91.99 | 0.831 / 0.977 |
| Dot Product | 34.56 | 72.05 / 91.17 | 73.65 / 91.99 | 0.834 / 0.985 |
| Penumbral$^\dagger$ | **35.56** | 72.67 / 91.12 | 74.34 / 92.38 | 0.835 / **0.990** |
| Umbral$^\dagger$ | 35.07 | **73.14 / 91.82** | **74.46 / 92.54** | **0.836** / 0.989 |
| $d_{\mathbb{H}}$ $\mathsf{H}^n$ $\xi$ | 32.54 | 49.19$^*$ / 74.68$^*$ | – | 0.834 / 0.987 |
| $d_{\mathbb{H}}$ Hyperboloid | 33.80 | 0.10$^*$ / 0.45$^*$ | – | 0.13$^*$ / 0.989 |
| Laplacian Kernel | 34.68 | 71.25 / 90.84 | – | 0.823 / 0.986 |

| Method | Adaptive Inputs WikiText-103 0 / 480 Context Window (Ppl. ↓) | Without Adaptive Inputs | DiT-B/4 @ 400K Steps (FID-50K ↓) |
|---|---|---|---|
| Dot Product | 20.86 / 19.22 | 26.62 / 24.73 | 68.9 |
| Penumbral | **20.72 / 19.01** | **26.44 / 24.31** | 67.7 |
| Umbral | 21.16 / 19.59 | 27.82 / 26.73 | **67.6** |

## 5 Results

Table 1 summarizes the performance of cone attention across various models and tasks. Both penumbral and umbral attention significantly outperform baselines on the NMT IWSLT and DeiT-Ti Imagenet tasks. For DeiT-Ti, umbral attention achieves 73.14% top-1 and 91.82% top-5 accuracy at 300 epochs and 74.46% top-1 and 92.54% top-5 accuracy at 500 epochs, which matches a distilled DeiT-Ti with more parameters (74.5% / 91.9%) [28]. On the GAT tasks, cone attention again outperforms baselines and achieves Graph Convolutional Network-level performance on the PPI dataset. Interestingly, almost all baselines, including dot product attention, outperform the concatenation-based attention in the original GAT paper. The two GAT tasks also reveal some interesting differences between penumbral and umbral attention. The Cora citation dataset is more tree-like with an average of 2 edges per node and chronological dependencies between nodes (a paper cannot cite a newer paper), while the PPI dataset is closer to a clustered graph with 14.3 edges per node [32, 13, 17]. As noted in [37], umbral cones appear to be better for strict hierarchies, while penumbral cones capture more complicated relationships such as those in the PPI dataset.

The adaptive inputs method regularizes models by reducing $d$ for rare words. We expect rarer words to be closer to the leaves of a word hierarchy, so the adaptive inputs method also acts as a hierarchical prior on the data [34]. We use adaptive inputs to test cone attention when combined with other hierarchical priors. Penumbral attention outperforms dot product attention with or without adaptive inputs, but the gap between the two methods is larger without adaptive inputs. On the other hand, umbral attention performs worse than dot product attention and penumbral attention on the WikiText103 task, with or without adaptive inputs. Since umbral cones are not geodesically convex, which means that the shortest path between two points in a cone may not necessarily lie entirely in that cone, we suspect that convexity is important for the WikiText-103 language modeling task. In the DiT model, patches are taken at the latent space-level, which we suspect gives less hierarchical information than when patches are taken from an input image [22]. Here, cone attention still outperforms dot product attention, but to a lesser degree. This mirrors our expectation that the DiT model does not benefit as much from hierarchical attention, and verifies that in such cases, using cone attention does not hurt performance. Interestingly, umbral cones slightly outperform penumbral cones in the DiT-B/4 task, which may be a result of patches being over the latent space, and not the input image.

### 5.1 Effect of Mapping Functions

Table 2 compares how $\psi$ and $\xi$ compare to the exponential map and psueodpolar map (section 3.2) on the NMT IWSLT task. Here, we use the exponential map at the origin of $\mathsf{H}^d$, $O = (0, 0, ..., 1)$. To compare $\mathrm{Exp}_O(v)$ to $\psi$, we first take $\mathrm{Exp}_O(v)$ and then project the point onto the boundary of

Table 2: Comparison of various mappings from Euclidean space to hyperbolic models for the NMT IWSLT task (BLEU scores, higher is better). The exponential map generally performs worse than $\psi$ and the pseudopolar map. While $\psi$ also performs worse than the psuedopolar map, table 1 indicates that the pseudopolar map is less numerically stable than $\psi$.

| Method | $\mathrm{Exp}_O(v) \to \mathsf{H}^d$ | $\xi \to \mathsf{H}^d$ | $\psi \to \mathsf{H}^d$ | Pseudopolar $\to$ Hyperboloid |
|---|---|---|---|---|
| Penumbral | 35.12 | **35.56** | - | - |
| Umbral | 34.63 | - | **35.07** | - |
| $d_{\mathbb{H}}$ | 30.59 | - | 32.54 | **33.80** |

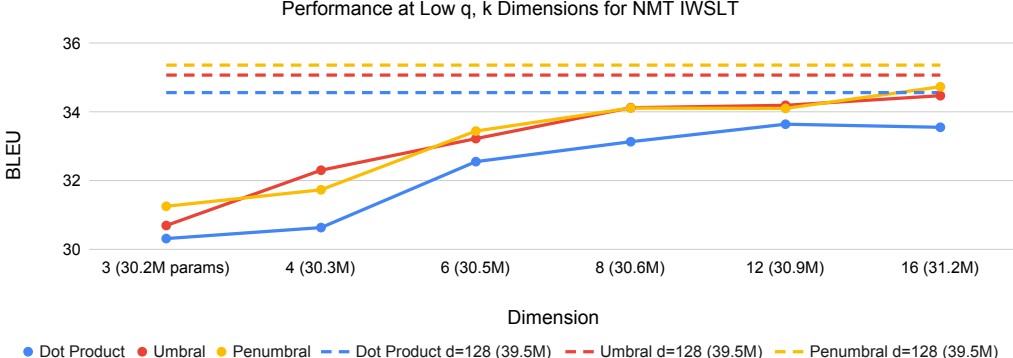

Figure 4: Performance of cone and dot product attention at low dimensions on NMT IWSLT. The base architecture uses 128 dimensions, and cone attention is able to match dot product attention with only 16 dimensions. For this model, this allows us to use 21% fewer parameters to reach performance parity, indicating that cone attention more efficiently captures hierarchical information.

$L$ if $\mathrm{Exp}_O(v)$ is in $L$. In the infinite penumbral setting, this corresponds to taking $\mathrm{Exp}_O(v)_d = \min(\mathrm{Exp}_O(v)_d, h)$. $\psi$ and $\xi$ generally perform much better than taking the exponential map at the origin $O$, which suggests that $\psi$ and $\xi$ have better optimization properties. For $d_{\mathbb{H}}$ attention, Gulcehre et al.'s pseudopolar map slightly outperforms $\xi$. However, table 1 indicates that outside of this specific task, using the psuedopolar map and Hyperboloid is less numerically stable.

### 5.2 Attention Efficiency and Model Size

Figure 4 shows the performance of cone attention vs. dot product attention at low *token* $(q, k)$ embedding dimensions ($d$ from before) on the NMT IWSLT task. Both umbral and penumbral attention are able to achieve significantly better performance than dot product attention in this regime, matching dot product attention at $d = 128$ with only $d = 16$. For this model, using 16 dimensions reduces the number of parameters from 39.5M to 31.2M. Table 3 shows the performance of DeiT-Ti at $d = 16$. Here, penumbral attention at $d = 16$ is able to outperform dot product attention at $d = 64$, again indicating that cone attention is able to more efficiently capture hierarchies.

### 5.3 Sensitivity to Initialization

Hyperbolic embeddings are known to be sensitive to initialization [10, 37]. To test cone attention's sensitivity to initialization, we trained 5 seeds for the IWSLT De2En task for cone attention and dot

Table 3: Performance of DeiT-Ti at 64 (default) and 16 dimensions, 300 epochs.

| Method | d = 64 | d = 16 |
|---|---|---|
| Dot Product | 72.05 / 91.17 | 71.29 / 90.54 |
| Penumbral | 72.67 / 91.12 | **72.25 / 91.20** |
| Umbral | **73.14 / 91.82** | 71.67 / 90.71 |

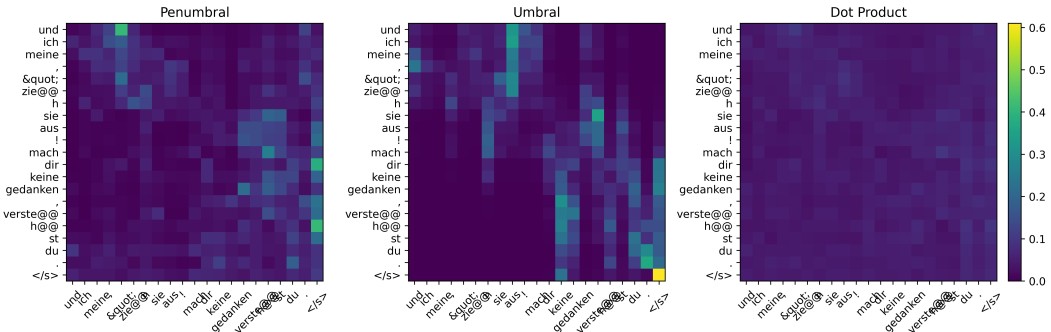

Figure 5: Attention heatmaps from an attention head in a trained IWSLT De2En translation model for the tokenized validation sequence *"Und ich meine, "Zieh sie aus! Mach dir keine gedanken, verstehst du.'*, which translates to *"I'm like, "Get it off! Don't worry about it, you know."* The cone attention heatmaps (left and center) have a clear distinction between the two parts of the sequence separated by "!", whereas the dot product heatmap (right) does not have a clear separation.

product attention. Dot product had an average BLEU score of 34.59 and standard deviation 0.12, penumbral achieved $35.41 \pm 0.14$, and umbral achieved $35.03 \pm 0.30$. There was one outlier in the umbral attention trials, and with the outlier removed umbral achieved $35.16 \pm 0.09$. The cone attention methods appear to have slightly higher variance than dot product attention, but not significantly so.

### 5.4   Attention Heatmaps

A natural question arises about what cone attention methods actually learn. Figure 5 shows heatmaps from an attention head in a trained IWSLT De2En translation model. The heatmaps for penumbral and umbral attention show clearer separation than the dot product attention heatmap. Furthermore, the separation in the two cone attention methods happens at the exclamation mark, a natural segmentation of the sequence into two parts. Intuitively, cone attention can be seen as an attention method that satisfies certain "logical constraints," such as "if $z \prec y$ and $y \prec x$, then $z \prec x$," which leads to relations between attention scores. For example. if $K(x, y)$ and $K(y, z)$ are both high, then $K(x, z)$ should also be relatively high in cone attention. In dot product attention, this is not guaranteed. If $x = (1, 1, 0, 0), y = (10, 10, 10, 10)$, and $z = (0, 0, 1, 1)$, then $\langle x, y \rangle = \langle y, z \rangle = 20$, but $\langle x, z \rangle = 0$. We suspect this is a reason why cone attention methods show better separation than dot product attention, which can aid performance.

## 6   Conclusion

We introduce cone attention, a *hierarchy-aware* method for calculating attention. Cone attention relies on entailment cones and the geometric properties of hyperbolic space to capture complex structural patterns that dot product attention does not explicitly model. We test cone attention in a variety of attention networks ranging from a few thousand to a few hundred million parameters, and achieve consistent performance improvements in NLP, vision, and graph prediction tasks over dot product attention and baselines. Cone attention also matches dot product attention with significantly fewer embedding dimensions, opening the potential for smaller models. These results suggest that cone attention is an effective way to encode hierarchical relationships in attention, and can potentially improve task-level performance in a wide variety of models and task domains.

**Future Work.** It remains to be seen how cone attention scales to very large models. Beyond this, [10] and [37] suggest that hyperbolic embeddings are sensitive to initialization, which implies that different transformer weight initializations may affect cone attention.

## Acknowledgements

This work was supported by NSF Award IIS-2008102. Compute resources were provided by the Cornell G2 Cluster.

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
