# 7 Appendix

## 7.1 Additional Hyperbolic Manifolds and Maps

### 7.1.1 Pseudopolar Coordinates and the Hyperboloid Model

The pseudopolar map $\pi$ used in [13] is given by

$$\pi(x)_{:-1} = x_{:-1} \sinh(x_d) \qquad \pi(x)_d = \cosh(x_d) \tag{13}$$

using the notation from the main text.

The hyperboloid model $\mathcal{H}^d$ is a model of $\mathbb{H}^d$ in the $d+1$ dimensional Minkowski space. Formally,

$$\mathcal{H}^d = \{x \in \mathbb{R}^{d+1} | \langle x, x \rangle_M = -1, x_{d+1} > 0\} \tag{14}$$

where

$$\langle q, k \rangle_M = \left( \sum_{i=1}^{n} q_i k_i \right) - q_{d+1} k_{d+1} \tag{15}$$

The distance metric on $\mathcal{H}^d$ is given by $d_{\mathbb{H}}(q, k) = \mathrm{arcosh}(-\langle q, k \rangle_M)$. We refer the reader to [13] and [4] for more information on the Hyperboloid model.

### 7.1.2 Klein Model

The Klein model used in [13] for Einstein aggregation is given by $\mathcal{K}^d = \{x \in \mathbb{R}^n | \|x\| < 1\}$. A point in $\mathcal{K}^d$ can be obtained from a point in $\mathcal{H}^d$ by taking the projection

$$\pi_{\mathcal{H}^d \to \mathcal{K}^d}(x)_{i \leq d} = \frac{x_i}{x_{d+1}} \tag{16}$$

with inverse

$$\pi_{\mathcal{K}^d \to \mathcal{H}^d}(x) = \frac{(x, 1)}{\sqrt{1 - \|x\|}} \tag{17}$$

The distance in $\mathcal{K}^d$ can be computed with $d_{\mathbb{H}}(q, k) = \mathrm{arcosh}(-\langle \pi_{\mathcal{K}^d \to \mathcal{H}^d}(q), \pi_{\mathcal{K}^d \to \mathcal{H}^d}(k) \rangle_M)$. We refer the reader to [13] and [4] for more information on the Klein model.

### 7.1.3 Poincaré Ball Model and Ganea's Cones

The Poincaré ball model is closely related to the Poincaré half-space, and is defined as the manifold $(\mathcal{B}^d, g_p)$, where $\mathcal{B}^d = \{x \in \mathbb{R}^d | \|x\| < 1\}$ and

$$g_p(x) = \left( \frac{2}{1 - \|x\|^2} \right)^2 g_e. \tag{18}$$

The distance between two points $q$ and $k$ on the Poincaré ball is given by

$$d_{\mathbb{H}}(q, k) = \mathrm{arcosh} \left( 1 + 2 \frac{\|q - k\|^2}{(1 - \|q\|^2)(1 - \|k\|)^2} \right) \tag{19}$$

When $\|x\| \to 1$, $d_{\mathbb{H}}$ changes very quickly, which makes optimization difficult near the boundary of the Poincaré ball.

Ganea's cones are defined by a radial angle function $\psi$. For a point $x$, the angle of the cone at $x$ is given by

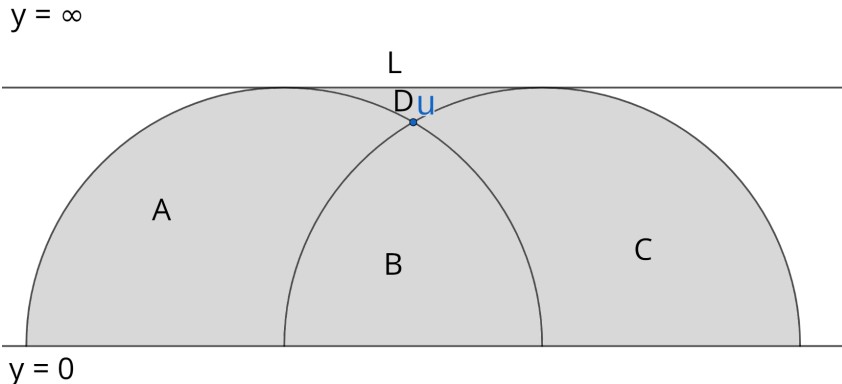

Figure 6: The gray area $(A \cup B \cup C \cup D)$ is the region of all points that share a cone with $u$. $B$ is the cone of $u$, and $D$ is the region of points whose cones contain $u$ (ancestors of $u$).

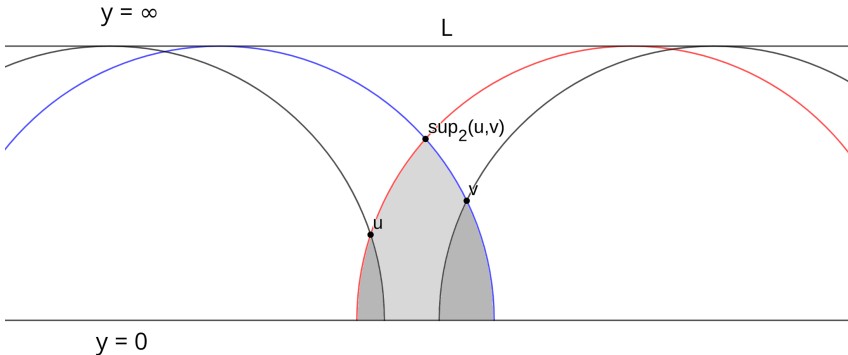

Figure 7: If $v_x \geq u_x$, then $\sup_2(u, v)$ is the intersection of the "right" geodesic of $u$ (red) and the "left" geodesic of $v$ (blue).

$$\psi(x) = \arcsin\left(K\frac{1 - \|x\|^2}{\|x\|}\right) \tag{20}$$

where $K$ is a user-chosen constant that defines the size of the $\epsilon$-ball. Clearly, cones do not exist for small $x$, giving the $\epsilon$-ball.

### 7.2 Penumbral Attention Derivation

Recall the definition of Penumbral Attention in the infinite setting shadow cone construction:

$$K(u,v) = \exp\left(-\gamma \max\left(u_d, v_d, \sqrt{r^2 - \left(\frac{\sqrt{r^2 - u_d^2} + \sqrt{r^2 - v_d^2} - \|u_{:-1} - v_{:-1}\|}{2}\right)^2}\right)\right) \tag{21}$$

when there exists a cone that contains $u$ and $v$, and

$$K(u,v) = \exp\left(-\gamma\sqrt{\left(\frac{\|u_{:-1} - v_{:-1}\|^2 + u_d^2 - v_d^2}{2\|u_{:-1} - v_{:-1}\|}\right)^2 + v_d^2}\right) \tag{22}$$

otherwise. There exists a cone that contains $u$ and $v$ when

$$\left( \| u_{:-1} - v_{:-1} \| - \sqrt{r^2 - u_d^2} \right)^2 + v_d^2 < r^2 \qquad \text{or} \qquad \| u_{:-1} - v_{:-1} \| \leq \sqrt{r^2 - u_d^2} \qquad (23)$$

Below, we give a derivation of this definition in the infinite penumbral cone construction. From theorem 2, $\sup_2(u, v)$ is the lowest cone that contains $u$ and $v$ in the plane containing $u, v$, and the ideal point of $\hat{\mathcal{S}}$. This lets us derive the height of $\sup_2(u, v)$ in that plane, which is given by the intersection of Euclidean semicircle geodesics with Euclidean radius $r$.

First, we check if there exists a cone that contains $u$ and $v$ in this plane. Note that the region of points $v$ s.t. $\exists$ a cone containing $u$ and $v$ is the region contained by the two geodesics forming the cone of $u$ (see figure 6). Assume $u_x = 0$. Then, in the infininte setting penumbral construction, these geodesics are the two semicircles centered at $\left( \pm \sqrt{r^2 - u_y^2}, 0 \right)$ with radius $r$. We can check if a point is in this region by checking if it is in the quarter circle bounded by $\left( -\sqrt{r^2 - u_y^2}, 0 \right)$ and the arc from $\left( -\sqrt{r^2 - u_y^2} - r, 0 \right)$ to $\left( -\sqrt{r^2 - u_y^2}, r \right)$, the quarter circle bounded by $\left( \sqrt{r^2 - u_y^2}, 0 \right)$ and the arc from $\left( \sqrt{r^2 - u_y^2} r, 0 \right)$ to $\left( \sqrt{r^2 - u_y^2}, r \right)$, or the rectangular region between the two. This is given by

$$\left( \left( \left( v_x - \sqrt{r^2 - u_y^2} \right)^2 + (v_y - 0)^2 < r^2 \right) \wedge v_x > \sqrt{r^2 - u_y^2} \right) \vee v_x \leq \sqrt{r^2 - u_y^2} \qquad (24)$$

which reduces to equation 23 since $u_x = 0 \implies v_x = \| u_{:-1} - v_{:-1} \|$.

Now, we derive the height of $\sup_2(u, v)$ when there exists a cone containing $u$ and $v$. Assume WLOG that $v_x \geq u_x$, and normalize $u$ and $v$ s.t. $u_x = -\| u_{:-1} - v_{:-1} \|/2$ and $v_x = \| u_{:-1} - v_{:-1} \|/2$. If $u \nprec v$ and $v \nprec u$, $\sup_2(u, v)$ is intersection of the "right" geodesic of $u$ and the "left" geodesic of $v$ (see figure 7). The center of the right geodesic of $u$ is given by $\left( u_x + \sqrt{r^2 - u_y^2}, 0 \right)$, and the center of the left geodesic of $v$ is given by $\left( v_x - \sqrt{r^2 - v_y^2} \right)$. Since $\sup_2(u, v)$ is equidistant from these two centers, $\sup_2(u, v)_x = \left( \sqrt{r^2 - u_y^2} - \sqrt{r^2 - v_y^2} \right)/2$. Then,

$$\sup_2(u, v)_y = \sqrt{ r^2 - \left( \frac{\sqrt{r^2 - u_y^2} - \sqrt{r^2 - v_y^2}}{2} - \left( u_x + \sqrt{r^2 - u_y^2} \right) \right)^2 } \qquad (25)$$

$$= \sqrt{ r^2 - \left( \frac{\| u_{:-1} - v_{:-1} \| - \sqrt{r^2 - u_y^2} - \sqrt{r^2 - v_y^2}}{2} \right)^2 } \qquad (26)$$

$$= \sqrt{ r^2 - \left( \frac{\sqrt{r^2 - u_y^2} + \sqrt{r^2 - v_y^2} - \| u_{:-1} - v_{:-1} \|}{2} \right)^2 } \qquad (27)$$

If $u \prec v$ or $v \prec u$, then $\sup_2(u, v)$ is equal to $u$ or $v$, respectively. In this situation, equation 27 is less than $u_y$ or $v_y$, repsectively. Thus, we take the max of $u_y, v_y$, and equation 27 to get equation 21.

When there does not exist a cone containing $u$ and $v$, we return the height of the lowest light source s.t. there exists such a cone. This corresponds to the height of the highest point in the extended geodesic through $u$ and $v$. Since geodesics are Euclidean semicircles, this height is the radius of the Euclidean semicircle through $u$ and $v$. Let the center of this semicircle be $(x, 0)$, and normalize $u$ and $v$ s.t. $u_x = 0$ and $v_x = \| u_{:-1} - v_{:-1} \|$. Then, we have

$$x^2 + u_y^2 = (v_x - x)^2 + v_y^2 \tag{28}$$

$$2v_x x = v_x^2 + v_y^2 - u_y^2 \tag{29}$$

$$2\|u_{:-1} - v_{:-1}\|x = \|u_{:-1} - v_{:-1}\|^2 + v_y^2 - u_y^2 \tag{30}$$

$$x = \frac{\|u_{:-1} - v_{:-1}\|^2 + v_y^2 - u_y^2}{2\|u_{:-1} - v_{:-1}\|} \tag{31}$$

The radius of the semicircle through $u$ and $v$ is then

$$\sqrt{\left(\frac{\|u_{:-1} - v_{:-1}\|^2 + v_y^2 - u_y^2}{2\|u_{:-1} - v_{:-1}\|}\right)^2 + u_d^2} \tag{32}$$

It is easy to verify that this is symmetric in $u$ and $v$ and gives equation 22. Furthermore, it is also easy to verify that when

$$\left(\|u_{:-1} - v_{:-1}\| - \sqrt{r^2 - u_d^2}\right)^2 + v_d^2 = r^2 \tag{33}$$

the in-cone and out-of-cone heights are both $r$, and equations 21 and 22 both equal $\exp(-\gamma r)$, making $K$ continuous with respect to the positions of $u$ and $v$.

## 7.3 Umbral Attention Derivation

Recall the definition of Umbral Attention in the infinite setting shadow cone construction:

$$K(u, v) = \exp\left(-\gamma \max\left(u_d, v_d, \frac{\|u_{:-1} - v_{:-1}\|}{2\sinh(r)} + \frac{u_d + v_d}{2}\right)\right) \tag{34}$$

As shown in [40], cone regions for infinite setting umbral cones are given by Euclidean triangles with angle apertures of $2\arctan(\sinh(r))$. WLOG, assume that $v_x \geq u_x$ and normalize $u$ and $v$ s.t. $u_x = 0$ and $v_x = \|u_{:-1} - v_{:-1}\|$. When $u \not\prec v$ and $v \not\prec u$, $\sup_2(u, v)$ is given by the intersection of the line with slope $1/\sinh(r)$ through $u$ and the line with slope $-1/\sinh(r)$ through $v$. This gives

$$y - u_y = \frac{x - u_x}{\sinh(r)} \tag{35}$$

$$y - v_y = \frac{x + v_x}{\sinh(r)} \tag{36}$$

$$2y = \frac{v_x - u_x}{\sinh(r)} + v_y + u_y \tag{37}$$

$$y = \frac{\|u_{:-1} - v_{:-1}\|}{2\sinh(r)} + \frac{v_y + u_y}{2} \tag{38}$$

where $\sup_2(u, v) = (x, y)$. When $u \prec v$ or $v \prec u$, equation 38 is less than $u_y$ or $v_y$, respectively. Thus, we take the max of $u_y, v_y$, and equation 38, giving us equation 34. When $v_x < u_x$, $\sup_2(u, v)$ becomes the intersection of the line with slope $1/\sinh(r)$ through $v$ and the line with slope $-1/\sinh(r)$ through $u$, which gives us the same result.

## 7.4 Theorems

**Theorem 1.** *In the infinite shadow cone construction, the cone with root farthest away from $\mathcal{S}$ that contains $u$ and $v$ is the minimum height cone that contains $u$ and $v$.*

*Proof.* We prove the umbral and penumbral cases separately.

*Penumbral Cones:* The distance from boundary of $\mathcal{S}$ to a point $x$ is the length of geodesic orthogonal to $\mathcal{S}$ through $x$. In the infinite penumbral construction, $\mathcal{S}$ is a horosphere, so this geodesic is the vertical Euclidean line from $x$ to $\mathcal{S}$. Clearly, the longer this line is, the lower $x$ is.

*Umbral Cones:* Here $\mathcal{S}$ is a point, and geodesics through $\mathcal{S}$ are vertical Euclidean lines orthogonal to the "$x$-axis" (using the definition of "$x$-axis" from the main text). As with the penumbral case, since the geodesic from a point $x$ to $\mathcal{S}$ is vertical Euclidean line, the longer this line is the lower $x$ is. $\quad\square$

**Theorem 2.** *In the infinite shadow cone construction, the root of the minimum height cone that contains $u$ and $v$ lies on the plane containing $u$, $v$, and $\mathcal{S}$ (or the ideal point of $\mathcal{S}$).*

*Proof.* We prove the umbral and penumbral cases separately.

*Penumbral Cones:* Consider the region of points $x$ s.t. $x \prec u$. This is the region of geodesics through $u$ that intersect $\mathcal{S}$ and is axially symmetric around the vertical Euclidean line $A_u$ through $u$ [40]. Denote this region $\mathcal{C}_u$ and its boundary $\mathcal{B}_u$. Note that the ideal point of $\mathcal{S}$, $x_d = \infty$, is the intersection of vertical line geodesics, so all planes through $u$ and $x_d = \infty$ contain $A_u$. Since $\mathcal{C}_u$ is axially symmetric w/r.t. $A_u$, it follows that it is reflection-symmetric across such planes.

Now, consider the intersection of $\mathcal{C}_u$ and $\mathcal{C}_v$, which has boundary $\mathcal{B}_{u \cap v}$. $\mathcal{C}_u \cap \mathcal{C}_v$ is clearly reflection-symmetric along the plane $P$ containing $u$, $v$, and $x_d = \infty$. As such, for any plane $P'$ orthogonal to $P$, all points on $\mathcal{B}_{u \cap d} \cap P'$ are either on $\mathcal{B}_u$ or $\mathcal{B}_v$ but not both. Since the geodesics that form $B_u$ are monotonically increasing in height in the direction away from $A_u$ (and likewise for $v$), the minimum height cone that contains $u$ and $v$ must have root in $\mathcal{B}_{u \cap v}$. Furthermore, the root of minimum height cone on $\mathcal{B}_{u \cap v} \cap P'$ is the closest point on $\mathcal{B}_{u \cap v} \cap P'$ to $A_u$. Since the set of closest points on a plane to a parallel line is the intersection of the orthogonal plane through the line, the minimum height cone on $\mathcal{B}_{u \cap v} \cap P'$ is on $P$. Thus, the minimum height cone that contains $u$ and $v$, which is the minimum over $P'$ of minimum height cones on $\mathcal{B}_{u \cap d} \cap P'$, is on $P$.

*Umbral Cones:* The proof for the umbral construction is identical to the penumbral construction, except that $\mathcal{S}$ is a point at $x_d = \infty$.

$\quad\square$

## 7.5 Implementation Details

All experiments were run on a shared GPU cluster with various machine configurations. All GPUs in the cluster were NVIDIA Volta or newer cards, and all machines had Intel Skylake or newer CPUs or AMD Milan or newer CPUs. Most experiments used PyTorch 2.0 with `torch.compile` and TF32 turned on for matrix multiplications. We provide PyTorch implementations of the infinite-setting penumbral and umbral attention operators at `https://github.com/tsengalb99/coneheads`. Below, we give implementation details for each tested model. For penumbral attention, we set the height of $\mathcal{S}$ to $h = 1$. For umbral attention, we set the radius of the ball around each point to $r = 0.1$

**Graph Attention Networks.** We use the Pytorch-GAT repository (`https://github.com/gordicaleksa/pytorch-GAT`) for our experiments. In this repository, we modified the `//models/definitions/GAT.py` file to implement various attention mechanisms. We use the provided training commands and hyperparameters to train models. We experimented with different hyperparameters, which did not have a significant effect on the final results. We report the default attention results from the original GAT paper.

**Neural Machine Translation (NMT) Transformers.** For NMT experiments, we used the fairseq repository available at `https://github.com/facebookresearch/fairseq`. We primarily modified `//fairseq/modules/multihead_attention.py`. We used the commands provided at `https://github.com/facebookresearch/fairseq/blob/main/examples/translation/README.md` to download and preprocess the IWSLT'14 De-En dataset and to train models.

**Vision Transformers.** For DeiT-Ti experiments, we use the Facebook DeiT repository available at `https://github.com/facebookresearch/deit/blob/main/README_deit.md`. We observed that all methods performed better at 500 epochs than at 300 epochs, and we report our experimental results for 300 and 500 epochs. The DeiT repository relies on the Hugging Face timm repository, which is available at `https://huggingface.co/timm`. We primarily modify `//models/vision_transformer.py` in the timm.

Table 4: Training speed in iterations per second of various attention models across the 4 tested transformers. When used in transformers, cone attention is slightly slower than dot product attention. As mentioned in the main text, `torch.compile` is not optimal at fusing operations, and the performance gap can likely be narrowed with a custom CUDA implementation. `torch.compile` was not used for the NMT transformer since the training speed was sufficiently fast and, as of writing, `torch.cdist` has issues with dynamic-shaped tensors. We expect this to be fixed in future PyTorch releases and the performance to be similar to the Adaptive Inputs transformer. All numbers were obtained on a single NVIDIA Tesla V100 SXM2 GPU.

| Method | NMT (4096 tokens/it) | Adaptive Inputs (4096 tokens/it) | DeiT-Ti (bs=256) | DiT-B/4 (bs=64) |
|---|---|---|---|---|
| Dot Product | 9.09 | 0.526 | 62.9 | 1.09 |
| Penumbral | 7.10 (-21.9%) | 0.490 (-6.86%) | 51.3 (-18.6%) | 1.08 (-0.92%) |
| Umbral | 7.48 (-17.7%) | 0.488 (-7.32%) | 56.0 (-11.0%) | 1.07 (-1.83%) |
| `torch.compile`? | No | Yes | Yes | Yes |

**Adaptive Input Representations for Transformers.** For adaptive inputs experiments, we use the same fairseq repository as the NMT experiments. The `transformer_lm_wiki103` architecture in the fairseq repository uses 8 attention heads per module, which does not match 16 attention heads per module in in [5]. Thus, our results and those from other papers that use this repository, such as [41], are not directly comparable to the results presented in [5].

**Diffusion Transformers.** For DiT-B/4 experiments, we use the code and training instructions available at `https://github.com/facebookresearch/DiT`. We use the Pytorch version of the codebase, and train with seed 42 to match the experiments presented in [25] and the repository. The DiT codebase also uses timm, so we make the same modifications as in DeiT. We report the DiT-B/4 seed 42 PyTorch result for dot product attention from the DiT Github repository.

## 7.6 Runtime Comparison

Table 4 shows the training speed of the 4 tested transformers in iterations per second. Like dot product attention, cone attention requres $O(n^2 d)$ operations. However, cone attention requires more operations, since dot product attention can be computed with a batch matrix multiply. Inside a transformer, cone attention generally results in a 10-20% performance decrease when implemented with `torch.compile`. However, `torch.compile` is not perfect, and an optimized raw CUDA implementation would likely narrow the speed gap between dot product and cone attention. `torch.compile` was not used for the NMT transformer since the training speed was sufficiently fast and, as of writing, `torch.cdist` has issues with dynamic-shaped tensors. We expect this to be fixed in future PyTorch releases and performance to be similar to the Adaptive Inputs transformer.

## 7.7 $\alpha$-approximate rank

Here, we discuss an interesting and potentially useful connection between the problem of encoding hierarchies in dot product attention and the $\alpha$-approximate rank problem. This connection gives us some intuition as why larger models with large token embedding dimensions perform well, even with hierarchical data. We note that we have not extensively studied this connection and how it relates to training dynamics, such as in the context of learning attention with gradient based optimizers, and leave that for future work.

Consider the following formulation of encoding a partial ordering in attention:

$$K(x,y) = \exp(\gamma P(x,y)) \qquad P(x,y) \in \begin{cases} [1, \alpha] & \text{if } x \preceq y \\ [-\alpha, -1] & \text{if } x \npreceq y \end{cases} \tag{39}$$

where $1 \leq \alpha \leq \infty$. Essentially, for a set of keys and a single query, the attention matrix should give higher "weight" to keys who are descendants of the query, which is similar to cone attention. If $\alpha$ is close to 1, then this gives a tighter margin on the attention matrix, which gives a "higher quality" attention after softmax normalization. We wish to characterize the number of dimensions $d$ needed in

dot product attention to encode a partial ordering with equation 39. Now, consider the $\alpha$-approximate rank of a sign matrix $A$:

**Definition 3.** $\alpha$*-approximate rank: For a sign matrix $A \in \{-1, +1\}^{n \times n}$, the $\alpha$-approximate rank is defined as the minimum rank of a matrix $A' \in \mathbb{R}^{n \times n}$ such that $J \leq A \circ A' \leq \alpha J$, where $J$ is the all one's matrix, $\alpha \geq 1$, and $\circ$ is the elementwise product.*

In dot product attention, $P = qk^\top$, where $q, k \in \mathbb{R}^{n \times d}$. Note that $P$ has rank $d$. Furthermore, if the partial ordering is encoded in a sign matrix $S$ s.t. $S_{ij} = +1$ if $i \preceq j$ and $-1$ otherwise, finding the minimum $d$ s.t. $\exists P$ that satisfies equation 39 reduces to finding the $\alpha$-approximate rank of $S$. A number of works have focused on characterizing the $\alpha$-approximate rank of arbitrary sign matrices. Below, we summarize some results from [2] and [17]. Lee and Shraibman [17] give the following bounds on the $\alpha$-approximate rank of a $n \times m$ sign matrix $A$, denoted $\mathrm{rk}_\alpha(A)$.

$$\frac{1}{\alpha^2}\gamma_2^\alpha(A)^2 \leq \mathrm{rk}_\alpha(A) \leq \frac{8192\alpha^6}{(\alpha-1)^6}\ln^3(4mn)\gamma_2^\alpha(A)^6 \tag{40}$$

where $\gamma_2^\alpha(A)$ is defined as

$$\gamma_2^\alpha(A) = \min_{A':J \leq A \circ A' \leq \alpha J} \gamma_2(A') \tag{41}$$

and $\gamma_2(A)$ is defined as

$$\gamma_2(A) = \max_{u,v:\|u\|=\|v\|=1} \|A \circ vu^\top\|_{tr}, \tag{42}$$

and

$$\mathrm{rk}_{\frac{\alpha+t}{1-t}}(A) \leq \frac{4\gamma_2^\alpha(A)^2\ln(4mn)}{t^2} \tag{43}$$

for $0 < t < 1$. These bounds depend on $\gamma_2^\alpha$, which, to the best of our knowledge, has not been well characterized for partial ordering matrices. However, $\gamma_2^\alpha$ can be computed in polynomial time with a semidefinite program, which may be useful [17]. Equation 40 gives some indication of how $\alpha$ changes with $d = \mathrm{rk}_\alpha$. From equation 40

$$\alpha \leq \frac{1}{1 - \sqrt[6]{\frac{8192\ln^3(4n^2)\gamma_2^\alpha(S)^6}{d}}} \tag{44}$$

For fixed $S$, as $d$ increases, this upper bound on the achievable $\alpha$ margin decreases with $O(1/(1 - \sqrt[6]{1/d}))$. Finally, if we go in the other direction and set $\alpha = \infty$, finding $d$ reduces to finding the sign-rank of $S$. From [3], the sign-rank of $S$ is bounded by $SC(S) + 1$, where $SC$ is the minimum over all column permutations of $S$ of the maximum number of sign changes in a row of $S$. If $S$ encodes a hierarchy, and the nodes of that hierarchy are numbered depth first in $S$, then $SC(S) \leq 2$, and so the sign-rank is at most 3.