# OpenReview forum: "Coneheads: Hierarchy Aware Attention"
_NeurIPS.cc/2023/Conference — NeurIPS 2023 poster_

### Official Review · Reviewer_F7eB · 2023-06-13

**Soundness:** 3 good
**Presentation:** 3 good
**Contribution:** 3 good
**Rating:** 6
**Confidence:** 2

**Summary:**

This paper introduces a simple drop-and-replace operation for dot-product attention, based on cones. The new operation is designed to model hierarchical information better than standard attention. The proposed method outperforms standard attention on various tasks. I will say upfront that I am not very familiar with the mathematical background used in this work, so my evaluation is mostly on the general framework and experiments.



**Strengths:**

- An interesting idea to present alternatives to the dot-product attention. Most existing works in this field target efficiency and simplicity, while this work is actually slower to run, but seems to improve performance. It also allows using smaller models and reaching similar performance, which can be thought of as a more efficient variant after all.
- Extensive experiments, showing that the proposed approach outperforms dot-product attention.



**Weaknesses:**

- The paper title is “Hierarchy Aware Attention”, and the mathematical model, to the best of my understanding, indeed promotes hierarchy. However, there is no explicit discussion about the desired hierarchy to be learned. Are the authors expecting the model, e.g., to learn the syntactic trees in sentences? The event graph in an image? Importantly, none of the experiments quantify this learned hierarchy either, which is a bit disappointing.

**Questions:**

- What are the model parameters? It would be nice to see a comparison between the proposed method and standard attention.

---

> ### Author Rebuttal · Authors · 2023-08-10
>
> - Hierarchical nature of model.
>     - Please see the general response. Beyond that, discovering hierarchical structures in an unsupervised or semi-supervised manner is an active research area, and it remains to be seen if cone attention can be applied there. Using cone attention for structure discovery is an interesting idea, and we leave that for future work.
> - “What are the model parameters? It would be nice to see a comparison between the proposed method and standard attention
>     - Figure 4 in the attached PDF shows the cosine between parameter tensors for 15 weight layers between dot product, penumbral, and umbral attention for the same seed. The cosines are all somewhat low, which suggests the parameter tensors are somewhat orthogonal and different. Furthermore, taking a trained dot product attention model and swapping out the attention calculation with cone attention (and vice versa) results in poor performance, which suggests the models are learning different things. For example, using dot product attention on a penumbral model gives 4.75 BLEU, and the other way around gives 3.62 BLEU

---

> > ### Comment · Reviewer_F7eB · 2023-08-21
> >
> > Thank you for the clarifications.
> >
> > "discovering hierarchical structures in an unsupervised or semi-supervised manner is an active research area, and it remains to be seen if cone attention can be applied there": I think that the argument that promoting hierarchy within attention leads to improved attention should be backed up by some evidence. Figs 1-3 in the attached pdf is a nice start, but might be anecdotal given that it's only one example. Showing a few more examples, or better yet, running syntactic probing experiments (e.g., https://aclanthology.org/N19-1419/) could be more convincing.

---

> > > ### Author Response · Authors · 2023-08-21
> > >
> > > Thank you for your suggestion regarding syntactic probing. This is an interesting idea that would indeed shed more light in a principled way on what hierarchical structures the model is learning. Given that the discussion period ends in a few hours, we will try to explore this direction for the camera ready version of the paper.

---

### Official Review · Reviewer_hcYy · 2023-06-25

**Soundness:** 3 good
**Presentation:** 3 good
**Contribution:** 3 good
**Rating:** 7
**Confidence:** 3

**Summary:**

The paper explores a variant of hyperbolic attention to induce a hierarchical inductive bias by exploiting the corollaries of points in a hyperbolic space and nodes in a tree. The paper integrates shadow cone construction for entailment cones to define partial order relations between points (the partial order relation can be interpreted as hierarchical relations in a tree structure) replacing a prior method from Ganea et a. 2018. The authors then model the similarity function of attention over any two vectors roughly based on the distance of the lowest common ancestor of them based on the partial ordering defined by shadow cone constructions. Intuitively this would lead to giving higher attention to representations that have close-by levels of hierarchy as per their locations in the hyperbolic space.

**Strengths:**

* Reasonable contribution in the space of hyperbolic embeddings/hyperbolic attention. Can be of interest to the community in that space.
* Fair improvement in different domains and over prior works like Gulcehre et al. 2018.
* Hyperbolic embeddings tend to be more effective with less dimensions. Improving in this space can be meaningful for development of smaller more effecient models - although there is a question if there is a diminishing return as we increase the dimensions.

**Weaknesses:**

* The limitation says that the model can be sensitive to initialization. While I am not going to take that itself as a negative point but this fact may put some concern given that the models are only trained once per data. However, since the improvement is quite consistent across multiple tasks and datasets (even if run only once), I think the concern is mitigated a bit. One potential way to show some more insights related to sensitivity to initialization within resource constraints - could be to try running multiple seeds in only a subset of tasks (maybe even just one of them) and show the comparisons with error bars.

(I have read the rebuttal)

**Questions:**

Will the code be open-sourced? Also, how is the empirical time/memory trade-off?

**Limitations:**

Limitations seem fairly addressed.

---

> ### Author Rebuttal · Authors · 2023-08-10
>
> - “The limitation says that the model can be sensitive to initialization. While I am not going to take that itself as a negative point but this fact may put some concern given that the models are only trained once per data. However, since the improvement is quite consistent across multiple tasks and datasets (even if run only once), I think the concern is mitigated a bit. One potential way to show some more insights related to sensitivity to initialization within resource constraints - could be to try running multiple seeds in only a subset of tasks (maybe even just one of them) and show the comparisons with error bars.”
>     - Hyperbolic embeddings have been noted to be sensitive to initialization in prior works, such as Ganea et al.’s entailment cones. As you noted, the empirical results suggest that cone attention may be relatively robust to initialization. We suspect this is because in most of our tasks, cone attention is used within a much larger model (e.g. a transformer) as opposed to traditional graph embedding tasks, where embeddings are directly optimized with few additional parameters.
>     - We ran an experiment where we trained the IWSLT De2En model with 5 different seeds to measure sensitivity to initialization. Below are the mean and standard deviation of the 5 trials. There was one outlier (34.51 BLEU) for umbral attention. The cone attention methods have slightly higher variance between runs than dot product attention, but not significantly so.
>
> Multiseed experiments
> - Dot product: mean 34.59 std 0.12
> - Penumbral: mean 35.41 std 0.14
> - Umbral with outlier: mean 35.03 std 0.30
> - Umbral without outlier: mean 35.16 std 0.09

---

> > ### Comment · Reviewer_hcYy · 2023-08-10
> > **Response**
> >
> > Thank you for the additional experiments and discussion.
> >
> > My concerns and questions are mostly satisfactorily addressed.

---

### Official Review · Reviewer_AaRb · 2023-07-06

**Soundness:** 3 good
**Presentation:** 3 good
**Contribution:** 3 good
**Rating:** 6
**Confidence:** 5

**Summary:**

This paper introduces hyperbolic shadow cone similarity and induced Hierarchy Aware Attention. The main contribution lies in their proposal to utilize hyperbolic shadow cones (specifically, umbral cone and penumbral cone) as an entailment setting to induce pairwise similarity scores denoted as $K(\cdot,\cdot)$ in hyperbolic space $\mathbb{H}^d$. By replacing the euclidean inner-product in Transformers with this hyperbolic similarity score based on shadow cones, the model achieves comparable performance across various tasks such as graph prediction, natural language processing (NLP), and vision tasks, while significantly reducing the number of parameters required.

**Strengths:**

The concept of utilizing shadow-cone based similarity and extending it to attention is innovative.

Applying this shadow-cone based hyperbolic similarity score to Transformers results in a notable reduction in the number of parameters required, while maintaining performance comparable to the Euclidean baselines.

The experiments conducted in this study are extensive, covering graph prediction, natural language processing (NLP), and vision tasks.

**Weaknesses:**

While entailment cones do enforce the encoding of hierarchical information, the learning scheme employed in this paper differs fundamentally from that of [12]. In [12], a contrastive loss embeds the hierarchy of data points, while an entailment loss ensures the preservation of partial order relationships. The extent to which this method is "hierarchical" is neither explicitly explored within the method nor through experiments.

The use of "Attention" in the title may be a bit over-claiming, as this paper primarily contributes to defining pairwise similarity within hyperbolic cones, rather than addressing other aspects of the attention mechanism. It is worth noting that in the closely related work [13] proposed an attention framework that covers various aspects, including the construction of K, Q, and V in hyperbolic space, as well as their operations.

The speed of calculating similarity is a crucial aspect of the attention mechanism, but it is only briefly mentioned in Section 3.1.

**Questions:**

As the word ``Hierarchical`` is in the title, can the authors clarify on why with this similarity mechanism, the model becomes ``Hierarchical`` some qualitative results on how the hierarchy is built?

What is the cost of using hyperbolic cones? More specifically, How much slower cone attention is compared with euclidean dot-product in different tasks over graph prediction, NLP, and vision domain?

**Limitations:**

The research is about fundamental mechanisms of machine learning models, so it does not have potential negative societal impact.

---

> ### Author Rebuttal · Authors · 2023-08-10
>
> - Why is the model hierarchical?
>     - We apologize if our use of “hierarchical” caused confusion and are open to using other terms in the final paper. Please see the general response for more information about hierarchies and the model.
> - The use of "Attention" in the title may be a bit over-claiming, as this paper primarily contributes to defining pairwise similarity within hyperbolic cones, rather than addressing other aspects of the attention mechanism. It is worth noting that in the closely related work [13] proposed an attention framework that covers various aspects, including the construction of K, Q, and V in hyperbolic space, as well as their operations.
>     - Thank you for your feedback, we are open to using other terms in the paper. However, we would like to point out that we do define multiple mappings ($\xi, \psi$) into hyperbolic space that are critical to our method. We did not explore hyperbolic aggregation as Gulcehre et al. did, since hyperbolic aggregation is independent of how similarity between data points is measured. However, our method is fully compatible with hyperbolic aggregation, which we leave for future work.

---

> > ### Comment · Reviewer_AaRb · 2023-08-10
> > **More clarification will be appreciated.**
> >
> > * In general response, the authors mentioned "Our method relies on the implicit hierarchy defined by hyperbolic entailment cones". How is it implicitly defined as the authors definition and training is fundamentally different from Ganea's entailment cone?
> > * In the PDF of general response, figures 1, 2, and 3 have different scales, we can hardly conclude that there is a "hierarchical" structure inside.
> > * In the supplementary material, table 5 measures the running speed, but the caption is inconsistent with the content. The caption mentioned that cone attention is slightly slower but the content shows the proposed cone attention runs faster, why is this case? It would be very interesting (and important) to show how much slower it is in the vanilla version.

---

> > > ### Author Response · Authors · 2023-08-10
> > >
> > > Thank you for your prompt response, we have written individual responses to the three questions below.
> > > - In general response, the authors mentioned "Our method relies on the implicit hierarchy defined by hyperbolic entailment cones". How is it implicitly defined as the authors definition and training is fundamentally different from Ganea's entailment cone?
> > >     - Our definition is consistent and compatible with Ganea et al.’s entailment cones, which are a special case of shadow cones. Ganea et al. define a hierarchy with entailment cones, which is used for partial order embedding tasks with ground truth partial orderings. In our work, we use the hierarchy defined by entailment cones to propose a similarity measure with LCAs in this hierarchy. Since in our tasks, we do not have access to ground truth hierarchical information, we use our formulation so as to learn potential implicit hierarchies (e.g. of tokens) in the latent space of the model. We hope this clarifies our motivation.
> > > - In the PDF of general response, figures 1, 2, and 3 have different scales, we can hardly conclude that there is a "hierarchical" structure inside
> > >     - Figures 1 (penumbral) and 2 (umbral) have scales from 0 to around 0.5. Figure 3 (dot product) has a scale of 0 to around 0.1. The post-softmax distributions learned in dot product attention are much closer to the uniform distribution, whereas cone attention learns a clear separation between the two sentences. If we had plotted all three on the same scale, dot product attention would essentially be all blue since the values are much more uniform than cone attention. The different scales were used so the plots had better visual dynamic range.
> > > - In the supplementary material, table 5 measures the running speed, but the caption is inconsistent with the content. The caption mentioned that cone attention is slightly slower but the content shows the proposed cone attention runs faster, why is this case? It would be very interesting (and important) to show how much slower it is in the vanilla version.
> > >     - In table 5, the units are **iterations per second**, as described in the caption. A lower number means the model trains slower. The percentages next to the numbers are to show how much lower the throughput of training a cone attention model is.

---

> > > > ### Comment · Reviewer_AaRb · 2023-08-17
> > > >
> > > > Regarding the frequently discussed "hierarchy" question, each of the three figures uses a different scale, making the comparison challenging. Since the authors are considering placing less emphasis on hierarchy in the next version, I regard this as resolved concern and won't delve further into this subject.
> > > >
> > > > Thank you for clarifying the running speed. It's somewhat surprising to learn that the reduced number of parameters does not lead to faster training, but considering a smaller model still has its significance. I have no further questions and will remain positive to this paper.

---

### Official Review · Reviewer_Rx82 · 2023-07-07

**Soundness:** 3 good
**Presentation:** 2 fair
**Contribution:** 4 excellent
**Rating:** 6
**Confidence:** 3

**Summary:**

This paper proposes replacing the traditional softmax attention kernel with one that tries to captures the hierarchical relationship between two vectors by mapping them to hyperbolic space.

**Strengths:**

The idea of using hyperbolic attention cones to define hierarchy-aware attention is (to my knowledge) novel in the literature. The specific choices of cone construction and mapping into hyperbolic space is also a contribution of the paper.

The numerical results appear quite strong, where the proposed attention method outperforms dot product attention in all settings, and retains better performance at lower dimensions.

**Weaknesses:**

While the paper validates the proposed attention method in terms of numerical results, one thing that's not studied is whether the behavior of the trained models actually matches the intuition that gave rise to the method, namely the use of hierarchy in the attention process. It would help complete the story if there were some results of this nature, such as attention heatmaps that exhibit hierarchical behavior, or a demonstration of how learned vectors induced by the attention mechanism correspond to meaningful tree structures. (It is unfortunately the case with neural networks that even when an idea works well, the *reason* it works may have little to nothing to do with the original motivation)

There are also certain aspects of the presentation that could be made more clear to the reader (or at least to me, see my questions below).

It might also be helpful to have a comparison with hyperbolic distance attention (Gulcehre et al) in a results table, to clarify whether the benefits come just from the use of hyperbolic geometry, or whether the proposed contributions related to entailment cones are in fact critical to achieving the reported results.

**Questions:**

There are a few details of the presentation that I couldn't quite understand.
- On line 191, you introduce $\xi$ as a way of satisfying the requirements of the penumbral setting, but in Table 2 $\xi$ only appears for the *Umbral* method. Likewise on line 190 $\psi$ is associated with the umbral construction, but in Table 2 $\psi$ is paired with *Penumbral*. Could you explain?
- Figure 2 showcases both finite setting and infinite setting entailment cones. What about the actual attention method and Definitions 1 & 2? Am I correct in understanding that only the infinite setting is being considered from that point of the paper onward?
- On lines 158/159, a user-defined quantity $r$ is introduced, which appears to be a hyperparameter. Is that right? I found no mention in the paper of how this hyperparameter is set or tuned. Could you say more on this, and whether choice of hyperparameter might have any impact on the relative performance of the umbral and penumbral methods?

The best-performing penumbral attention method appears to have a piece-wise definition. One regime is when an entailment cone exists that contains both u and v with respect to a fixed light source, and the other regime is when the cone does not exist. Have you checked which regime is actually used by your trained models? I could see several possibilities here:
- Perhaps after training attention mostly falls into equation (5), and the piecewise definition is only needed to avoid issues during optimization
- Or maybe a trained model mostly falls under equation (6)? In that case, wouldn't it be possible to dispense with the piecewise definition and use equation (6) as the entirety of the attention mechanism? Have you tried this?
- Or perhaps the boundary criterion (7) is being used as a critical source of nonlinearity when comparing attention queries and keys?
Knowing this would help me make sense of whether the actual operation of the attention method is in line with the proposed intuition of respecting hierarchy.

Are there any plans to release code? The paper is unfortunately not very accessible to neural network practitioners unfamiliar with hyperbolic embeddings and related literature, and having drop-in code would be helpful to allow the broader community to try out the proposed approach.


**Limitations:**

No concerns regarding limitations.

---

> ### Author Rebuttal · Authors · 2023-08-10
>
> - Hierarchical behavior of trained models
>     - Please see the general response.
> - Comparison with hyperbolic distance attention, hyperbolic geometry, and entailment cones
>     - We suspect that most of the performance difference between the two methods comes from our use of entailment cones vs. Gulcehre et al.’s use of hyperbolic distance. Both hyperbolic distance attention with the hyperboloid map and the $\xi$ map failed to converge on multiple tasks, which suggests that hyperbolic distance attention is less robust during optimization. The hyperbolic distance between two points grows exponentially with respect to the distance between the points and the hyperbolic origin, which may be the reason for these instabilities. Table 2 in the paper shows a comparison between the two attention methods under various mappings into hyperbolic space. The mapping and hyperbolic model used can also influence the end result, so this may be another factor in the performance difference as well.
> - $\xi$ and $\psi$
>     - Table 2 in the paper should read $\xi$ for penumbral and $\psi$ for umbral. This is a typo and we will fix it in future versions.
> - Infinite setting cones in the paper?
>     - Yes, from that point on everything uses infinite setting cones.
> - How to tune or set hyperparameters?
>     - We used h=1 for penumbral attention and r=0.1 for umbral attention. For some of the experiments, we experimented with different values of h and r but did not see any significant differences between runs. Using a higher h for penumbral attention reduces the chance that a random initialization falls into equation (6), but this may not actually matter much for a converged model. Using a lower h for a fixed $\gamma$ (temperature) increases the lowest similarity score achievable by penumbral attention with equation (5).
> - Piecewise definition of penumbral attention
>     - With an appropriately set combination of h and $\gamma$ (i.e. h$\gamma$ is not too low), points fall into both equations (5) and (6). We have not tried using only equation (6) as the attention kernel, which is an interesting idea. The boundary condition in equation (7) is to ensure piecewise definition is continuous. Using a different definition of equation (6) would also be interesting. We leave these directions for future work.

---

> > ### Comment · Reviewer_Rx82 · 2023-08-19
> >
> > Thank you for your answers and responses.
> >
> > The questions I have regarding piecewise definitions continue to concern me a bit, especially given that points fall under both regimes in a trained model. In that case, the simple story given in the paper (that equation 6 exists to prevent domain issues during optimization) doesn't seem to match the actual trained models. I worry that the piecewise behavior might effectively increase the depth of the network (by having the piecewise function act like another layer of nonlinearity), and the resulting higher expressiveness explains the better performance without involving the core motivation of hierarchy in hyperbolic space. I would suggest some further investigation here.
> >
> > The attention heatmaps show some hints of hierarchical structure emerging in the attention distributions, but overall the paper as a whole still doesn't leave me with a crystal clear picture of *how* the proposed method achieves better results that dot product attention. I also wonder whether sentence separation would emerge in the dot-product setting combined with rotary position embeddings (or perhaps ALiBi). My hunch is that the way these methods incorporate position information into the attention structure can help create similar sentence-separation patterns. I don't think this weakness of the paper is worth rejecting over, but as a reviewer I would still like to note that it hasn't been fully addressed.

---

> > > ### Author Response · Authors · 2023-08-20
> > >
> > > Thank you for your response and questions.
> > >
> > > > The questions I have regarding piecewise definitions continue to concern me a bit, especially given that points fall under both regimes in a trained model ...
> > >
> > > To clarify, without equation 6, penumbral attention would not exist for points whose LCA cone intersects the light source. The exact equation used in equation 6 is not necessarily set in stone. We chose that specific equation because it makes the penumbral attention function continuous. However, there are many other functions that would also make penumbral attention continuous. We have not explored using other functions for equation 6, but that is an interesting direction that we leave for future work.
> > >
> > > We are not sure we fully understand your statement that the “piecewise behavior might effectively increase the depth of the network.” Could you clarify this? We would like to point out that umbral attention is not piecewise but still outperforms dot product attention in most experiments, which suggests that a piecewise similarity score is not necessary for better performance. We also note that the laplacian kernel is a nonlinear similarity score, which suggests that a nonlinear similarity score is not sufficient for better performance.
> > >
> > > > The attention heatmaps show some hints of hierarchical structure emerging in the attention distributions, but overall the paper as a whole still doesn't leave me with a crystal clear picture of how the proposed method achieves better results that dot product attention ...
> > >
> > > This is an interesting question. Cone attention enforces a “soft logical constraint” on the attention in the sense that if we have three points $x, y, z$ such that $x \prec y$ and $y \prec z$ in the partial ordering imposed by cones, then $x \prec z$. Translated to attention scores, this means $K(x, z) < K(y, z)$. We have not investigated whether positional encodings such as rotary embeddings and ALiBi achieve the same final clustering effect on attention heatmap, but that is an interesting research direction that we leave for future work.

---

### Author Rebuttal · Authors · 2023-08-10

Dear reviewers, thank you for your detailed and thorough reviews. As reviewers Rx82 and AaRb noted, our use of entailment cones to define an attention mapping is “novel” and “innovative.” All reviewers noted that throughout our “extensive” suite of experiments across graph prediction, NLP, and vision tasks, cone attention demonstrated empirical improvements over dot product and prior hyperbolic attention works. Furthermore, reviewers hcYy and F7eB noted that cone attention matches dot product attention with fewer parameters, allowing for “smaller more efficient models.” Below, we have written responses to common questions about our paper. Namely, we have included a clarification about how hierarchies are used by cone attention. For responses to individual questions, please see our individual responses.

Beyond this, after the submission deadline and during the review period, we discovered an interesting result with our DeiT-Ti baseline. Unlike previously reported in the original DeiT-Ti paper, we found that DeiT-Ti did not plateau, and instead kept getting better with more epochs. In the version of the paper submitted for review, we reported the DeiT-Ti numbers at 300 epochs from the original paper, and reported penumbral attention numbers at 500 epochs, as we found that penumbral attention got better with more epochs. For transparency, we have compiled numbers from our DeiT-Ti experiments below for dot product and cone attention methods at 300 and 500 epochs. We note that penumbral and umbral attention both still outperform dot product attention at both 300 and 500 epochs, so our final conclusions are still the same. We will update Table 1 with the numbers from both 300 and 500 epochs. We also re-ran the experiments in Table 3, and the updated results are below. All methods do better than previously reported, and penumbral attention at d=16 now actually outperforms dot product attention at d=64. At d=16, penumbral and umbral attention still outperform dot product attention, and our conclusions do not change.

We are letting you know this updated information now in case this affects your reviews. We stress that cone attention still outperforms dot product attention for the DeiT-Ti task, and that our conclusions are still the same.

DeiT-Ti Table 1 update
- 300 epochs (top 1 acc / top 5 acc)
    - Dot Product: 72.05 / 91.17
    - Penumbral: 72.67 / 91.12
    - Umbral: **73.14 / 91.82**
- 500 epochs (top 1 acc / top 5 acc)
    - Dot Product: 73.65 / 91.99
    - Penumbral: **74.34 / 92.38**
    - Umbral: Still running, will be updated in a few days.



DeiT-Ti Table 3 update
- d = 64 (top 1 acc / top 5 acc)
    - Dot Product: 72.05 / 91.17
    - Penumbral: 72.67 / 91.12
    - Umbral: **73.14 / 91.82**
- d = 16 (top 1 acc / top 5 acc)
    - Dot Product: 71.292 / 90.542
    - Penumbral: **72.250 / 91.190**
    - Umbral: 71.668 / 90.714

## Responses to Common Questions

- Hierarchical nature of the model:
    - Our method relies on the implicit hierarchy defined by hyperbolic entailment cones to measure the similarity between two points. It does not take into account explicit ground truth hierarchies between points, such as a manually labeled syntax tree. Furthermore, such ground truth information was not available for the tasks we tested on. Essentially, we are learning a latent hierarchy in which the observed data points are only a subset of the nodes.
    - Figures 1-3 in the attached PDF show heatmaps from an attention head in a trained IWSLT De2En translation model for the tokenized sequence from the validation set ['und', 'ich', 'meine', ',', '&quot;', 'zie@@', 'h', 'sie', 'aus', '!', 'mach', 'dir', 'keine', 'gedanken', ',', 'verste@@', 'h@@', 'st', 'du', '.', '</s>'], which translates to ['i', '&apos;m', 'like', ',', '&quot;', 'get', 'it', 'off', '!', 'don', '&apos;t', 'worry', 'about', 'it', ',', 'you', 'know', '.', '</s>']. Figure 1 is from penumbral attention, figure 2 is from umbral attention, and figure 3 is from dot product attention. The cone attention heatmaps have a clear separation between the two sentences in the sequence (separated by “!”), whereas the dot product heatmap does not have a clear separation. This separation can be considered a hierarchy between the two parts of the sequence.
    - Furthermore, cone attention can be seen as an attention formulation that satisfies certain “logical constraints,” such as “if $z \prec y$ and $y \prec z$, then $z \prec x$," which leads to relations between attention scores. For example, if $K(x, y)$ and $K(y, z)$ are both high, then $K(x, z)$ should also be relatively high in cone attention. In dot product attention, this is not guaranteed. For example, if a = [1, 1, 0, 0], b = [10, 10, 10, 10], and c = [0, 0, 1, 1], <a, b> = <b, c> = 20 but <a, c> = 0. We suspect this is a reason why cone attention methods show better separation than dot product attention, which can aid performance.
- Will the code be open sourced?
    - Yes
- What is the runtime of cone attention?
    - Table 5 in the supplementary section has a simple benchmark on most of the tasks we ran experiments on. In general, using torch compile / writing an optimized kernel significantly reduces the runtime gap between cone attention and dot product attention.

---

### Author Response · Authors · 2023-08-16

From the main rebuttal:

> Umbral: Still running, will be updated in a few days.

Umbral attention achieves 74.46% top-1 accuracy and 92.54% top-5 accuracy when run for 500 epochs.

---

### Decision · Program_Chairs · 2023-09-21

**Decision:**

Accept (poster)

**Comment:**

Reviewers seem to be happy with this paper and there seems to be healthy discussions. Authors make an effort to run new experiments and improve the comprehensiveness of the work. This work looks kind of interesting though and perhaps a little novel. I recommend acceptance.